

# GnRH or estradiol benzoate combination with CIDR improves *in-vivo* embryo production in bovines (*Bos indicus* and *Bos taurus*) under subtropics

Khalid Mahmood[1], Muhammad Zahid Tahir[1], Mahboob Ahmad Butt[2], Shazia Mansoor Qureshi[3] and Amjad Riaz[1]

[1] Department of Theriogenology, University of Veterinary and Animals Sciences, Lahore, Punjab, Pakistan
[2] Department of Livestock Management, University of Veterinary and Animal Sciences, Lahore, Punjab, Pakistan
[3] Center of Excellence for Bovine Genetics, Okara, Punjab, Pakistan

Corresponding author
Amjad Riaz, dramjadriaz@uvas.edu.pk

## ABSTRACT

Multiple Ovulation and Embryo Transfer (MOET) technology is a potential technique to upgrade livestock species' genetics. The varied response to super-stimulatory treatments remains one of the limiting factors to this technology's widespread use. The present study was aimed to improve the superovulation response and *in-vivo* embryo production by using controlled internal drug release (CIDR)-GnRH or CIDR-EB (Estradiol Benzoate) along with conventional superovulation protocol in Holstein Frisian (HF): *Bos taurus; n = 42*) and Crossbred (XB: Cholistani (*Bos indicus*) × HF; $n = 28$) cows. In the CIDR-GnRH/CIDR-EB treatment, CIDR was implanted in the cows after confirming the presence of a corpus luteum (CL) on the 8th day after estrus. 2 ml GnRH (Lecirelin acetate 0.0262 mg/ml) or 2 mg EB was also administered in CIDR-GnRH/CIDR-EB groups, respectively. Both groups were given super-stimulatory treatment from the 11th day after estrus (FSH in tapering doses twice a day for four consecutive days). On day 13, two doses of 2 ml prostaglandin (75 µg/ml of dextrorotatory cloprostenol) were administered (am: pm), and CIDR was removed the following day. Two artificial inseminations (AI) of the cows were performed (12 h apart) on the 15th day. No CIDR and GnRH/E.B were given in the control group, but the remaining superovulation protocol was the same. Later on, seven days after the first AI, non-surgical embryo flushing was done. The transferable embryos produced from three different superovulation protocols were then transferred into the recipient cows ($n = 90$) for determining their fertility. Statistical analysis revealed that the number of super-estrus follicles (SEF), multiple corpora lutea (MCL), ovulation/fertilization percentage, fertilized structures recovered (FSR), and transferable embryos (TEs) remained significantly higher ($p < 0.05$), and days taken for return to estrus (RTE) after embryo collection remained significantly lower ($p < 0.05$) in CIDR-GnRH ($n = 18$) and CIDR-EB ($n = 15$) groups as compared to the control ($n = 37$). The comparison between XB and HF cows revealed that the TEs production in CIDR-GnRH (XB = 5 *vs* HF = 13) and CIDR-EB (XB = 6 *vs* HF = 9) based superovulation protocols were 11.60 ± 4.08 *vs* 04.31 ± 0.98 and 09.33 ± 1.78 *vs* 05.22 ± 1.36, respectively. TEs production in XB cows ($n = 5$) of the CIDR-GnRH group was significantly higher (11.60 ± 4.08) than other groups. On the other hand, the days taken for RTE after

embryo collection remained significantly lower ($p < 0.05$) in HF cows of treatment groups. However, the fertility of TEs was neither affected significantly ($p > 0.05$) by the superovulation protocol used nor by breed differences among donor cows. In conclusion, using CIDR-GnRH or CIDR-EB along with conventional superovulation protocol may enhance the efficiency of MOET programs in cattle. Furthermore, XB donor cows demonstrated a better performance than HF donor cows under subtropical conditions.

## INTRODUCTION

In the developing countries, over time increase in population has significantly increased the demand for animal source products such as meat and milk (*Mekonnen & Hoekstra, 2012*). The key to successfully meet the increasing demand for livestock products is to enhance productivity through improved genetic gain (*Lawanson & Oduntan, 2020*). The desire to improve animal genetics can be achieved using reproductive biotechnologies such as AI and MOET (*Wheeler et al., 2010*; *Naranjo-Chacón et al., 2019*). Generally, crossbreeding is considered an effective tool for quick genetic improvement of low milk-producing cows (*Galukande et al., 2013*). However, commercial dairy production in developing countries cannot sustain economically with F1 cows from nondescript animals due to low milk production potential.

Raising a purebred stock of dairy cows to produce F1 cows is confronted with constraints, such as the unavailability of the required number of native cattle and national breeding policies (*Leroy et al., 2016*). However, the propagation of high milk-producing readily available XB and HF cows through the MOET program to produce replacement heifers is a practical model (*Baruselli et al., 2018*; *Jarvis, 2019*; *Madalena, 2008*).

For developing countries like Pakistan, where most of the population is, directly and indirectly, dependent on agriculture, it becomes imperative to evaluate and disseminate reproductive biotechnologies-based methods to enhance animal productivity. *Warriach et al. (2015)* revealed the significance and the widespread dissemination of artificial insemination techniques in Pakistan. However, this study pointed out that the embryo transfer technique has yet to significantly succeed in Pakistan. *Mebratu, Fesseha & Goa (2020)* highlighted that the non-availability of the technical staff, the high cost involved in IVF/MOET, and the lack of fixed timeline-based superovulation protocols in subtropical conditions are the underlying factors for ET's low adaptability in developing countries.

In recent times, considerable advancement has been made in controlling ovarian function and biochemistry of gonadotropins; however, the inherent factors affecting superovulation outcomes in donor animals are still partially understood (*Bó & Mapletoft, 2014*; *Mapletoft, 2018*; *Phillips & Jahnke, 2016*; *Viana et al., 2018*). The preferred hormone for super-stimulation is FSH (follicle-stimulating hormone), as animals treated with eCG

(equine chorion gonadotropin) show abnormal LH profile and declined ovulation and fertilization rates (*Murphy, 2018*).

Traditionally, ovaries are super-stimulated during mid-cycle, *i.e.*, 8–12 days after estrus, because in cows with two follicular waves, an emergence of the 2nd follicular wave and the cohort of growing follicles is expected during mid-cycle (*Shehab El-Din, Abdel-Khalek & Bakr 2010*). However, this phenomenon differs in individual cows with three follicular waves (*Mapletoft, Steward & Adams, 2002*; *Adams & Singh, 2021*; *Hassan et al., 2021*). The 2nd follicular wave emerges one to two days earlier in cows with three follicular waves. Meanwhile, treatment with gonadotrophin on the day of follicle wave emergence results in a better super-stimulatory response (*Mapletoft & Bó, 2018*).

The ability to selectively induce the emergence of follicular waves allows super-stimulation to be initiated regardless of the estrous cycle stage and without the need to detect estrus and wait for the mid-cycle to start hormonal therapy (*Bo et al., 2002*; *Bó & Mapletoft, 2014*; *Jahnke & Youngs, 2021*). Previously, progesterone, estradiol, or GnRH has been used for elective emergence of follicular waves with remarkable success (*Bó et al., 2009*; *Guerrero et al., 2009*; *Nasser et al., 2011*; *Rivera et al., 2011*). Furthermore, the initiation of superovulation at the time of follicular wave emergence improves embryo production in cattle (*Bó & Mapletoft, 2014*; *Mikkola, Hasler & Taponen, 2020*). A recent study from Mexico has reported improved embryo production from XB dairy cows through a modified superovulation protocol in tropical conditions (*Chacón et al., 2020*). However, no specific superovulation protocol with promising results for exotic (*Bos Taurus*) and XB (*Bos taurus × Bos indicus*) cows under subtropical conditions has been reported from Pakistan. Therefore, the present study is an attempt to bridge this research gap by evaluating the superovulation response in the Pakistani-raised cows. The research was carried out to determine the optimal superovulation response and in-vivo embryo production in bovines by using CIDR-GnRH or CIDR-EB for synchronizing follicular wave before the initiation of superovulation in Holstein Frisian (HF: *Bos taurus; n* $= 42$) and Crossbred (XB: Cholistani (*Bos indicus*) ×HF; $n = 28$) cows under subtropical environment prevalent in Pakistan.

## MATERIALS AND METHODS

### Animal selection and ethical issues

A total of 70 donor cows (HF $= 42$ and XB $= 28$) with proper vaccination history and high milk production were super-stimulated in this study. All of the selected cows were of relative parity (1–3), age (3–8 years), and BCS more than 2.5 (*Roche et al., 2009*). The corresponding Committee for Animal Care, CEBG-Okara (QPY-5.3-01), approved all procedures involving animals' welfare and ethics. The Guiding Principles for the Care and Use of Laboratory Animals (*Baumans, 2005*; *Naderi et al., 2012*) were strictly adhered to in this research work.

### Management practices

The study was conducted at the Centre of Excellence for Bovine Genetics (CEBG), Okara, Punjab, Pakistan, located at N30°48′29″, E73°26′45″. To avoid heat stress, experiments

were carried out during moderate to colder months (October to March) with an average daily temperature of 22 °C. The annual average rainfall in the study area was 509 mm, and the average daily temperature ranged between 3 °C (in December) to 47 °C (in June/July). All sampled cows were housed in the open sheds, and their deworming was performed 30 days prior to the commencement of this study. All sampled cows were offered fresh green fodder/maize silage and concentrate with free access to fresh drinking water. Any animal showing abnormal vaginal discharge at the time of heat or history of recent illness was not selected for this study.

## Superovulation of donor animals

Cows in the natural estrus were subjected to three superovulation treatments, *i.e.,* CIDR-GnRH based, CIDR-EB based, and conventional superovulation. Donor cows of relative parity and age were randomly selected for these treatments. In the CIDR-GnRH group ($n = 18$) animals, the presence of CL was confirmed on Day 8 of the estrus cycle through ultrasonography (*Bényei et al., 2006*; *Kayacik et al., 2006*), and 2 ml GnRH (Dalmarelin; FATRO, Italy) was administered along with intravaginal placement of 1.38 g of progesterone CIDR (Controlled Internal Drug Release; Pfizer, USA). From Day 11–14, 400 mg of FSH (Folltropin-V; Bioniche Animal Health; Canada) was administered in 8 tapering (70 mg, 60 mg, 40 mg, and 30 mg) doses (AM: PM). On Day 13, 2 ml prostaglandin (Dalmazin; FATRO–Italy) was administered twice (AM: PM), followed by removal of CIDR on Day 14. On day 15, administration of 2 ml GnRH and AI was performed. The cows in super-estrus were inseminated twice (12 h apart) after confirmation of heat signs (*Roelofs et al., 2010*) with commercially available semen of HF bulls of proven fertility. The embryos were collected through non-surgical flushing on Day 22 (7 days post-AI), and all the animals received 2 ml of prostaglandin on Day 25 to induce luteolysis (Fig. 1A).

In the CIDR-EB group ($n = 15$), the animals were injected with 2 mg Estradiol Benzoate USP (Sigma Aldrich, USA) instead of GnRH. The remaining timeline was the same as described in the CIDR-GnRH group (Fig. 1B).

In the conventional (control) superovulation group ($n = 37$), the animals were not treated with CIDR and GnRH / EB, but the remaining superovulation timeline was the same as described in treatment groups (Fig. 1C). All of the procedures were performed by a qualified team of two veterinarians and three technicians.

## Ultrasound scanning

Each animal's ovaries were scanned using a 7.5 MHz linear rectal probe (iMAGO-S ECM, France) to check the existence of ovulatory follicle and CL on "Day 0" and "Day 8" of the estrous cycle, respectively. The scanning was continued on day 10, 11, and 12 to determine the pattern of follicular wave development and to rule out the presence of the dominant follicle. Any animal having a persistent dominant follicle till day 11 was not given FSH treatment. The ovarian scanning of donor animals was also performed on the day of super-estrus (before AI) and on the day of embryo collection. This was to determine the superovulation response and to count the total number of CLs present on the superovulated ovaries.

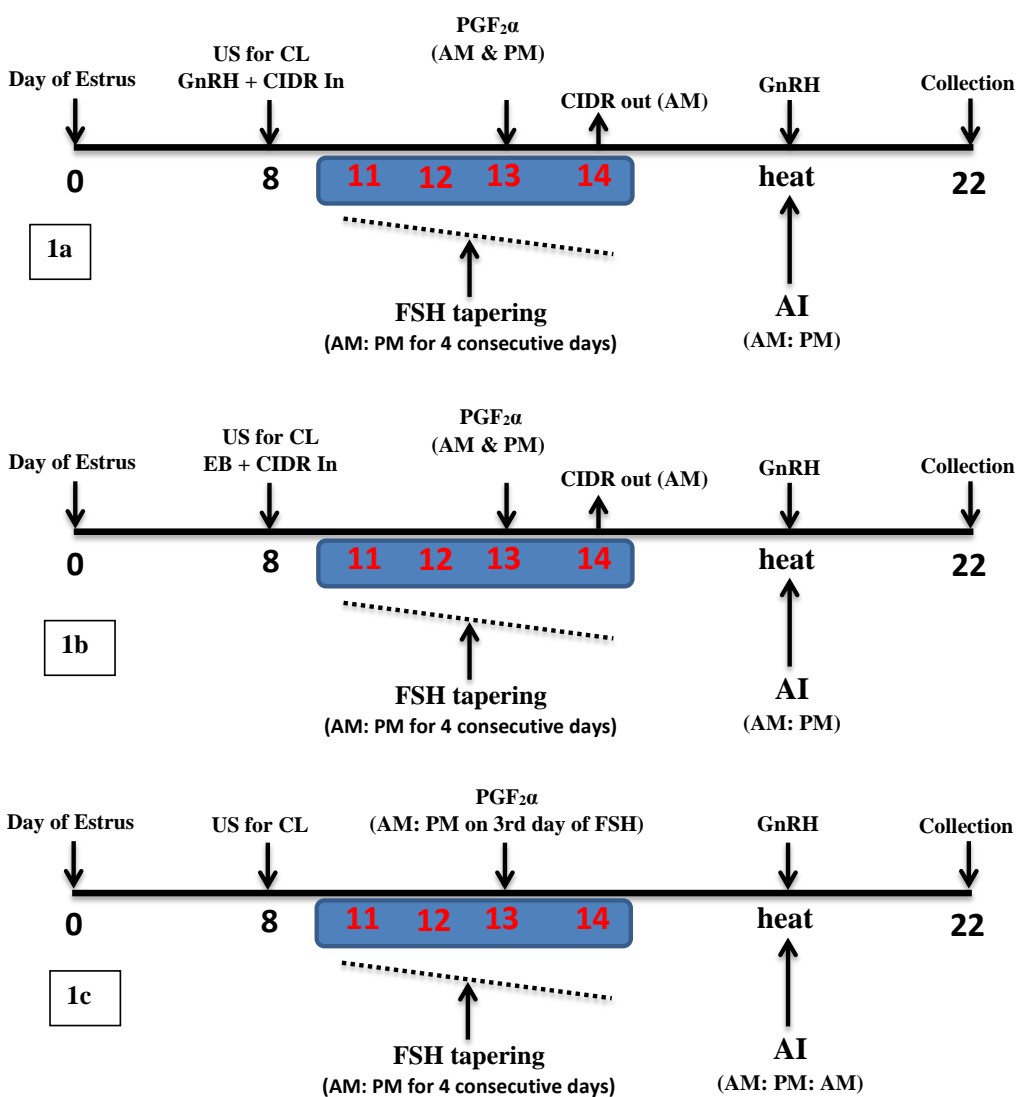

**Figure 1  Schematic diagram of three superovulation protocols for dairy cows.** (1A): CIDR-GnRH protocol, (1B): CIDR-EB Protocol, (1C): Conventional or Standard protocol. Day 0: heat (Natural estrus), US: ultrasonography, CL: Corpus luteum, FSH: follicle-stimulating hormone, EB: estradiol benzoate, AI: Artificial insemination, Collection: non-surgical embryo collection through flushing.

## Embryo flushing and grading

Embryos were collected from the donor animals using 18–24 gauge two-way Foley end Silicone catheter and flushing tubing after epidural anesthesia. Each horn was flushed using 480 ml of flushing media (Boviflush; Mintub, Germany), and the flushing fluid was filtered *via* 75 μmesh (Miniflush, Mintub, Germany). The mucus attached to the membranous part of the embryo filter was washed into a 64 mm petri dish using two-piece syringes and flushing media. Embryos were searched under 12× or 25× magnification of stereo zoom microscope and transferred into the holding media (Bovihold, Mintub, Germany). Embryos were washed (with holding media) up to eight times to remove any

debris before evaluating their quality under 50× magnification based on the regularity of shape, cellular mass, compactness of blastomere, the shape of zona pellucida (ZP), and the presence of extruded cells. The classification of the embryo was done based on its stage of development. Embryos having more than 85%, 50–85%, less than 50% compact cellular mass were classified as A, B, and C grade embryos, respectively (*Bó & Mapletoft, 2018*). Only A and B-grade embryos were considered for calculating TEs per donor per collection.

### Fertility trial of embryos

A total of 200 cow heifers of unidentified genotype or low-producing XB primiparous cows were used as ET recipients. The selected animals were observed for natural estrus, and ET was performed on the 7th day after standing heat. The presence of a good quality CL (*Bényei et al., 2006*; *Kayacik et al., 2006*) was confirmed through rectal palpation/ultrasonography before ET. Embryos (15 from HF and 15 from XB) from each treatment group (Total = 90) were randomly allotted to the recipient cows through the lottery method. ET was performed non-surgically (*Selk, 2002*), and pregnancy was checked by ultrasonography on the 45th day after estrus to calculate the fertility results.

### Study variables

The data was collected through ultrasonography for the ovarian picture of donor cows. The ovulation and presence of CL were checked on the day of estrus (Day "0"), Day "8" respectively. The number of multiple follicles (Super-Estrus Follicles, *i.e.,* SEFs) was checked on the day of super estrus. SEFs depict the success of FSH to recruit multiple follicles. The total number of multiple CLs (MCL) presented on the superovulated ovaries of each donor cow was checked on the day of embryo collection. MCLs indicate the successful ovulation of SEFs. All structures after non-surgical flushing were regarded as Total Structures Recovered (TSR). TSR were further dived into fertilized structures (Transferable Embryos, *i.e.,* "TEs" and Degenerated Embryos, *i.e.,* DGs) and unfertilized ova, *i.e.,* UFO (*Racowsky et al., 2010*). The days taken by the animals to return to estrus (RTE) were also observed. Ovulation percentages and fertilization percentages for every donor cow were calculated based on MCLs to SEFs ratio and FSR to TSR, respectively. The recipient cows showing a viable fetus on ultrasonography (38 days) after embryo transfer were considered pregnant. The pregnant recipients to total embryos transferred ratio was used to calculate the fertility percentage (*Tadesse et al., 2016*).

### Statistical analysis

All collected data was analyzed using SPSS 26.0. The normality of the data was checked using the Shapiro–Wilk test. The effects of a given treatment on ovarian structures (*i.e.,* SEFs and MCLs), TSR, FSR, TEs, DGs, UFO, days taken for RTE after embryo collection, ovulation percentage, and fertilization percentage were analyzed by Kruskal-Wallis test (non-parametric). Missing values for dependent variables were treated as missing without considering any case exclusion. Embryo fertility data was analyzed by the Chi-square method. Resultant graphs were generated using GraphPad Prism 8.2. All variations with a 95% confidence interval ($P < 0.05$) were treated as significant. Results have been presented as mean ± standard error of the mean (SEM).
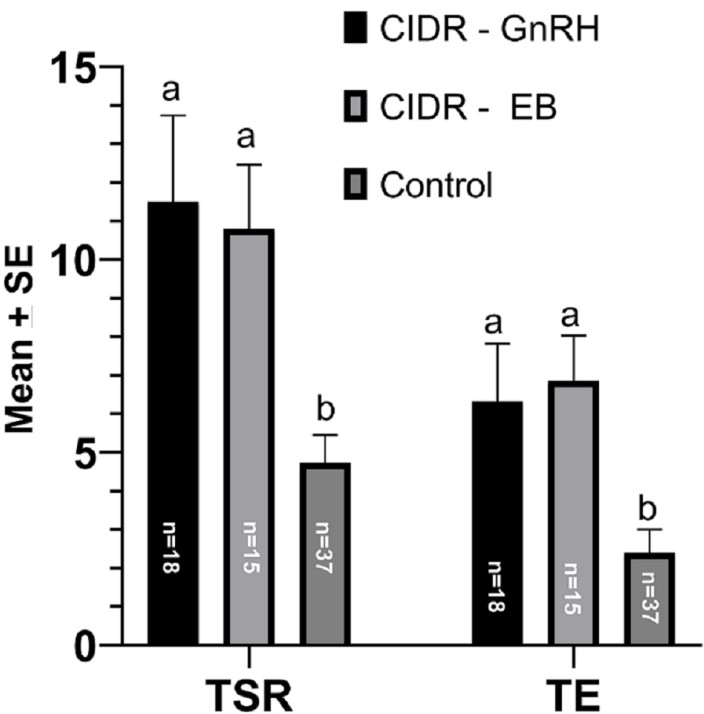

**Figure 2** *Mean (+ SEM)* **number of total structures recovered (TSR) and transferable embryos (TEs) in three different superovulation protocols.** AB Means without a common superscript differed ($P < 0.05$) Ref to Fig. 1 for treatment protocols.

## RESULTS

### Embryo production

The total structures recovered (TSR) per collection remained significantly higher ($P < 0.05$) in CIDR-GnRH (11.50 ± 2.24) and CIDR-EB (10.80 ± 1.66) groups as compared to the control (04.72 ± 0.73). Similarly, the number of transferable embryos per collection was significantly higher ($P < 0.05$) in CIDR-GnRH (6.33 ± 1.49) and CIDR-EB (6.86 ± 1.17) groups as compared to the control (2.40 ± 0.60). The results have been summarized in Fig. 2.

### Super-stimulation, ovulation, and fertilization

The mean number of super estrus follicles (SEF), ovulation percentages of these follicles, and percentage of fertilized structures recovered (FSR) remained significantly higher ($P < 0.05$) for CIDR-GnRH and CIDR-EB groups as compared to the control (Table 1).

### Return to estrus

After embryo collection, the mean number of days taken for return to estrus (RTE) were found significantly lower ($P < 0.05$) for both treatment groups as compared to the the control group. The mean value of RTE days for the CIDR-GnRH, CIDR-EB, and control group was 14.55 ± 2.14, 14.07 ± 1.74, and 22.86 ± 2.27, respectively.

**Table 1 Effect of CIDR-GnRH or CIDR-EB based superovulation protocols on ovulation and fertilization percentage (mean ± SEM).**

| Protocol | Follicles (Number) | Ovulation percentage (%) | Fertilization percentage (%) | UFO percentage (%) |
|---|---|---|---|---|
| **CIDR - GnRH** ($n = 18$) | 14.39 ± 1.63[a] | 68.17 ± 4.11[a] | 96.14 ± 1.96[a] | 3.86 ± 1.96[a] |
| **CIDR - EB** ($n = 15$) | 13.73 ± 1.31[a] | 77.79 ± 3.76[a] | 96.03 ± 1.96[a] | 3.97 ± 1.96[a] |
| **Control** ($n = 37$) | 08.00 ± 0.63[b] | 58.50 ± 4.84[b] | 57.60 ± 6.62[b] | 28.88 ± 5.75[b] |

**Notes.**

Refer to Fig. 1 for treatment protocol.

Follicles (large follicles at super-estrus), Ovulation percentage (percentage of super-estrus follicles which ovulated to form CLs on day of embryo collection), Fertilization percentage (percentage of fertilized structures recovered (FSR), *i.e.,* transferable and degenerated embryos out of total structures recovered), UFO (unfertilized ova), UFO percentage (percentage of UFO out of TSR).

[a,b]Within a column means without a common superscript differed ($P < 0.05$).

## Breed effects

As discussed above, the overall trend of super-stimulation, multiple ovulation, fertilization, embryo production, and RTE days remained the same when breed differences (*i.e.,* XB and HF) were considered. However, XB cows showed comparatively better results for both treatment groups' super-stimulation and embryo production parameters ($P < 0.05$). However, days taken for RTE remained significantly ($P < 0.05$) lower for HF cows. A graphical illustration of these results has been shown in Fig. 3.

## Embryo fertility results

Overall fertility percentage of embryo transfer trial remained 56.67 ± 5.25. On statistical analysis, the fertility of TEs was neither affected significantly ($P > 0.05$) by the superovulation protocol used nor by breed differences among the donor cows (Table 2).

## DISCUSSION

In the dairy industry, increased TEs production per donor per collection is the most acknowledged criterion for measuring the effectiveness of any bovine superovulation protocol. This study revealed that TEs production by HF and XB donor cows was significantly increased with modified superovulation protocols under a subtropical environment. The limited application of bovine embryo transfer technology in the developing countries of the subtropical environment is attributed to a lack of technological know-how and the inadequate super-stimulatory response of native breeds (*Sheetal, Prasad & Gupta, 2018*). The development of non-surgical ET techniques has provided an opportunity to propagate the genetic potential of superior females. Integration of MOET in superior bovine females can help in the genetic improvement of existing dairy herds (*Khan, 2002*). Due to the low milk production of domestic livestock breeds, Pakistan s introduced HF and Jersey Cattle to improve productivity (*Lateef et al., 2008*). However, these breeds can only be reared at controlled sheds due to heat stress-related problems and

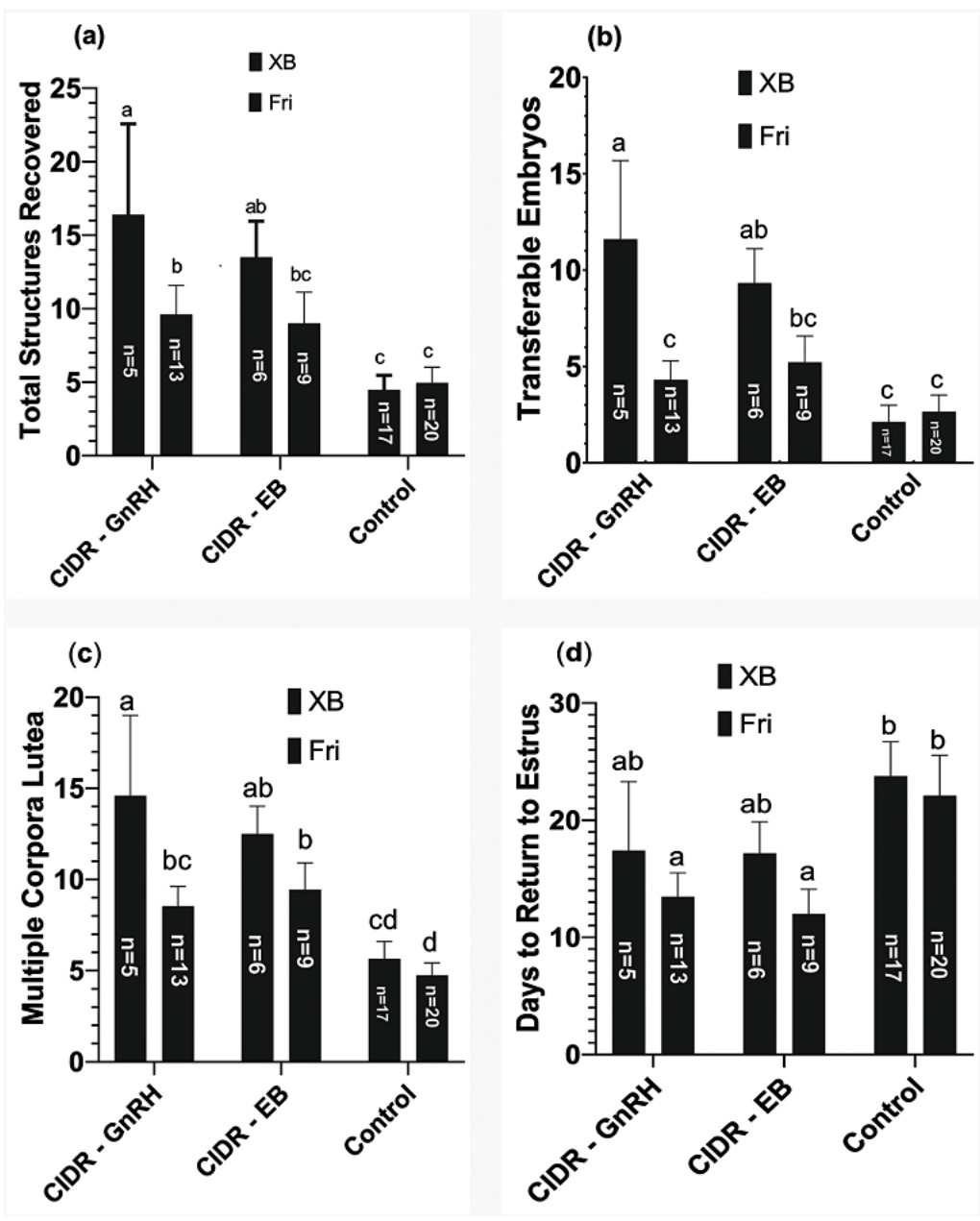

**Figure 3  Comparative results of three different superovulation protocols in Crossbred (XB) and Friesian (Fri) cows.** (A) Total structures recovered, (B) transferable embryos, (C) Multiple Corpora lutea, (D) days to return to estrus. [a–c]Means without a common superscript differed ($P < 0.05$). Refer to Fig. 1 for treatment protocols.

increased management costs. As a result, adversely impacting the overall dairy production economics (*Tahir et al., 2019*).

Crossbreeding of non-descript local cows with European cattle breeds is the best choice, as they have greater environmental variability adaptation and demonstrate a higher

**Table 2  Effects of protocols/breed differences on embryo fertility (mean ± SEM).**

| Protocol | Breed | Fertility percentage |
|---|---|---|
| CIDR - GnRH | XB ($n = 15$) | 9/15 (60.00%)[a] |
| | HF ($n = 15$) | 7/15 (46.67%)[a] |
| CIDR –EB | XB ($n = 15$) | 9/15 (60.00%)[a] |
| | HF ($n = 13$) | 7/13 (53.84%)[a] |
| Control | XB ($n = 15$) | 10/15 (66.67%)[a] |
| | HF ($n = 17$) | 9/17 (52.94%)[a] |

Notes.

Refer to Fig. 1 for treatment protocol. XB (Cholistani × HF crossbred cows), HF (Holstein Frisien Cows).

[ab] Within a column means without a common superscript differed ($P < 0.05$).

production performance (*Hassan & Khan, 2013*; *Leroy et al., 2016*). Furthermore, the economically viable application of MOET protocols in elite XB cows can accelerate genetic gain and improve livestock production. (*Granleese et al., 2015*). Using the methodology described in 'Materials and Methods' and applied in the subtropical Pakistani environments with extreme climatic variations revealed an increased number of transferable embryos per donor per collection in HF and XB cows. Moreover, the results of this study highlighted the efficiency of three super-stimulatory treatments subjected to HF and XB cows.

Increased understanding of the bovine estrous cycle's follicular wave dynamics has stimulated the dairy industry's interests for precise control of follicle/corpus luteal dynamics and ovulation timing (*Abdelnaby, Abo El-Maaty & El-Badry, 2020*). Follicular wave formation can be controlled mechanically or synchronized by hormonal therapy. A follicular ablation by an ultrasound-guided needle is performed for mechanical control, while for hormonal synchrony, GnRH or estradiol combined with progesterone is recommended (*Mikkola, Hasler & Taponen, 2020*). It is worth noting that CIDR's application supplemented with GnRH or EB prior to the initiation of super-stimulatory treatment results in new follicular wave emergence and yields a higher number of TEs than conventional super-ovulatory protocols (*Jahnke & Youngs, 2021*). The results of this intensive study have demonstrated a great prospect of induced follicular wave emergence treatments combined with super-ovulatory conventions. A similar superovulation study in Nelore (beef cows) conducted in Brazil has produced comparable results (*Andrade et al., 2003*).

Traditionally, 80% of the cows exhibit the emergence of two follicular waves in such type of experiments, whereas less than 20% showed three follicular waves (*Noseir, 2003*). Therefore, superovulation protocols are designed according to the day of the 2nd follicle wave emergence (9–12th of the estrous cycle) (*Mapletoft, Steward & Adams, 2002*). Synchrony between the time of follicular wave emergence and the commencement of bovine superovulation protocols results in improved response, whereas asynchrony (even for a day) negatively affects the *in vivo* embryo production (*Kara & Bekyürek, 2021*). In our study, the follicular wave's emergence was electively controlled using CIDR-GnRH or CIDR-EB. Afterward, superovulation treatment was initiated at the expected time of follicular wave emergence.

The main objective of any super-ovulatory program is to produce the maximum number of TEs per donor per collection. In our study, super-stimulatory response (*i.e.,* number of SEF at the time of super estrus and multiple CLs present on the day of embryo flushing) and in-vivo embryo production (TEs) improved with the CIDR-GnRH or CIDR-EB based superovulation treatment as compared to the control. These results were closer to the global mean TE production in dairy cows, *i.e.,* 5 per collection (*Bó & Mapletoft, 2014*). However, the degenerated embryo production remained significantly higher (4.61 ± 1.26 and 3.26 ± 0.63 and 1.16 ± 0.29, respectively) for treatment groups compared to the control group. This may be due to the higher serum P4 values during super-estrus due to the CIDR application in the presence of previous cycle CL. Other researchers have reported similar phenomena (*Wiley et al., 2019*; *Kara & Bekyürek, 2021*). Therefore, it is suggested to analyze these facts in correlation to serum P4 values. A similar study on Korean native cattle (*Bos Taurus*) has reported no difference in treatment and conventional group (*Son et al., 2007*). The range of TE among the different groups was 4–6 per donor per collection. Another study on Nelore (beef) cows has produced a higher mean number of TE with CIDR-GnRH treatment without considering the estrous cycle stage (*Andrade et al., 2003*). However, the TEs production in beef cows is generally higher as compared to dairy cows. Another study on Bos Taurus × Bos indicus (5/8 Holstein and 3/8 Zebu, and 5/8 Brown swiss × 3/8 zebu) multiparous cows has reported more than seven TEs per donor per collection (*Chacón et al., 2020*). Another study from Egypt about using CIDR and EB in superovulation protocol has reported 4.75 TEs per donor in HF from the best performing group (*Abdelkhalek et al., 2014*). However, the superovulation hormone used in this study was eCG. The results of our study were comparable to TE production from the middle-aged XB cow super-stimulated with CIDR-GnRH based superovulation under tropical conditions (*Naranjo-Chacón et al., 2019*). However, the TE production in treatment groups of our study was better than TE production from older XB cows (*Naranjo-Chacón et al., 2019*).

In hypothetical correspondence with studies which state that GnRH or porcine LH (pLH) administration results in ablation of a dominant follicle followed by the emergence of a new follicular wave 48–72 h later (*Macmillan & Thatcher, 1991*; *Pursley, Mee & Wiltbank, 1995*; *Martinez et al., 1999*; *Thatcher et al., 1993*; *ERDEM et al., 2020*), superovulation treatment was initiated on day 11 of the estrous cycle in our study.

Both treatment groups exhibited a similar wave emergence pattern (10.80 ± 0.20 days) and (10.33 ± 0.21 days), respectively. However, GnRH may fail to ablate large follicles, resulting in non-synchronization of follicle wave emergence (*Martinez et al., 1999*; *Guanga et al., 2020*). To address this phenomenon, estradiol-17β has been experimented on progestin-implanted cows. Such studies have reported a new follicular wave within 3 to 5 days (*Honparkhe et al., 2014*; *Singh et al., 2015*). Although estradiol-17β is a short-acting drug, its use on a commercial scale is impossible in many countries due to legal restrictions. Therefore, other commercially available estrogen esters (*i.e.,* estradiol benzoate or estradiol valerate) have also been experimented with, resulting in synchronous emergence of a new follicular wave within 3–4 days (*Bo et al., 2002*).

The two most important factors affecting the variability in superovulation results are the stage of the estrous cycle (follicular status of ovaries) and antral follicular count in donor cows (*Bó, Cedeño & Mapletoft, 2019*). Based on the findings of this study and perusal of the literature, it can be inferred that treating donor cows (for synchronization of follicular wave emergence) before initiating FSH administration remains beneficial in enhancing *in vivo* TEs production. Furthermore, the additional cost for the use of CIDR and GnRH/EB was much lower than its impact to improve the TEs production per donor per collection. Hence, the experimented timeline can benefit the dairy farmers by reducing the uncertainty about the suitable day (of estrous cycle) to start FSH treatment for producing better results.

Under field conditions, it may not be easy to perform ultrasound scanning and follow every cow to determine the exact day of follicular wave development. Hence a fixed timeline for superovulation and embryo production needs to be implemented.

The time interval between two consecutive embryo collections and the future utility of a superovulated cow is affected by the number of days taken by the animal for return to estrus (RTE) after embryo collection. Previously, it has been established that gap of two regular estrous cycles as a breeding rest is necessary (*Bó & Mapletoft, 2014*). Nevertheless, animals are being collected repeatedly with a reduced time interval (*Hasler, 2010*). The cows in the present study exhibited first heat $17.33 \pm 3.50$ days and $16.07 \pm 3.36$ days after embryo collection in CIDR-GnRH and CIDR-EB groups. Days taken by these groups for RTE were significantly lower ($P < 0.05$) than the control group, which may be correlated to a significantly higher ($P < 0.05$) ovulation rate in donor cows belonging to the treatment group. During the study, it was a general observation that cows with unruptured or un-ovulated follicles (on the day of embryo collection) took more days for RTE. However, more significant variations in the animals' days to come into first heat after collection need to be further explored. The donors are superovulated after every 30 to 40 days interval by many practitioners (*Hasler, 2014*). Reducing the days taken for RTE after embryo collection may help to achieve more TEs per donor per unit time (*Bó & Mapletoft, 2014*).

The fertility of embryos based on ET trials was not affected by the variation in super-ovulatory treatments given and the breed of the donor cows. Similar studies have reported no adverse effects on embryo fertility results in dairy or beef cows (*Andrade et al., 2002*; *Kara & Bekyürek, 2021*).

## CONCLUSION

It can be divulged from the present study that the addition of CIDR-GnRH or CIDR-EB treatment before traditional superovulation protocol can improve *in-vivo* embryo production in XB and HF cows under subtropical conditions without compromising the embryo fertility. Furthermore, XB donor cows can be a better candidate for MOET programs under subtropical conditions.

## ACKNOWLEDGEMENTS

The authors are very thankful to the technical and other staff working at the Centre of Excellence for Bovine Genetics, Okara, for their help in conducting this study.

### Funding

The Centre of Excellence for Bovine Genetics, Okara, Pakistan, provided the resources for this work. There was no additional external funding received for this study. The funders had no role in study design, data collection and analysis, decision to publish, or preparation of the manuscript.

### Grant Disclosures

The following grant information was disclosed by the authors:
The Centre of Excellence for Bovine Genetics, Okara, Pakistan.

### Competing Interests

The authors declare there are no competing interests.

### Author Contributions

- Khalid Mahmood conceived and designed the experiments, performed the experiments, analyzed the data, prepared figures and/or tables, authored or reviewed drafts of the paper, and approved the final draft.
- Muhammad Zahid Tahir analyzed the data, prepared figures and/or tables, authored or reviewed drafts of the paper, and approved the final draft.
- Mahboob Ahmad Butt conceived and designed the experiments, performed the experiments, prepared figures and/or tables, and approved the final draft.
- Shazia Mansoor Qureshi performed the experiments, analyzed the data, authored or reviewed drafts of the paper, and approved the final draft.
- Amjad Riaz conceived and designed the experiments, analyzed the data, prepared figures and/or tables, authored or reviewed drafts of the paper, and approved the final draft.

### Animal Ethics

The following information was supplied relating to ethical approvals (i.e., approving body and any reference numbers):

All procedures involving animals' welfare were approved by the Committee for Animal Care, Centre of Excellence for Bovine Genetics, Pakistan (QPY-5.3-01) and were performed following the Guiding Principles for the Care and Use of Laboratory Animals.

### Data Availability

The data that support the findings of this study are available in the Supplemental Files.

### Supplemental Information

Supplemental information for this article can be found online at http://dx.doi.org/10.7717/peerj.12077#supplemental-information.

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

# PeerJ

*of Faculty of Veterinary Medicine, Erciyes University/Erciyes üniversitesi Veteriner Fakültesi Dergisi* 18.

**Kayacik V, Salmanoğlu MR, Polat B, Özlüer A. 2006.** Evaluation of the corpus luteum size throughout the cycle by ultrasonography and progesterone assay in cows. *Turkish Journal of Veterinary and Animal Sciences* **29**:1311–1316.

**Khan M. 2002.** Prospects of multiple ovulation and embryo transfer (MOET) in genetic improvement of buffaloes. *Pakistan Veterinary Journal* **22**:35–39.

**Lateef M, Gondal K, Younas M, Sarwar M, Mustafa M, Bashir M. 2008.** Milk production potential of purebred Holstein Friesian and Jersey cows in subtropical environment of Pakistan. *Pakistan Veterinary Journal* **28**:9.

**Lawanson A, Oduntan O. 2020.** Emerging trends in animal reproductive technology– a review. In: *Proceedings of the 2nd international conference, the federal polytechnic, Ilaro, 10th–11th Nov. 2020. Available at* http://eprints.federalpolyilaro.edu.ng/id/eprint/1239.

**Leroy G, Baumung R, Boettcher P, Scherf B, Hoffmann I. 2016.** Sustainability of crossbreeding in developing countries; definitely not like crossing a meadow. *Animal* **10**:262–273 DOI 10.1017/S175173111500213X.

**Macmillan KL, Thatcher WW. 1991.** Effects of an agonist of gonadotropin-releasing hormone on ovarian follicles in cattle. *Biology of Reproduction* **45**:883–889 DOI 10.1095/biolreprod45.6.883.

**Madalena F. 2008.** How sustainable are the breeding programs of the global main stream dairy breeds?-The Latin-American situation. *Total Health* **138**:62.

**Mapletoft R. 2018.** History and perspectives on bovine embryo transfer. *Animal Reproduction* **10**:168–173.

**Mapletoft R, Bó G. 2018.** Innovative strategies for superovulation in cattle. *Animal Reproduction* **10**:174–179.

**Mapletoft RJ, Steward KB, Adams GP. 2002.** Recent advances in the superovulation in cattle. *Reproduction Nutrition Development* **42**:601–611 DOI 10.1051/rnd:2002046.

**Martinez MF, Adams GP, Bergfelt DR, Kastelic JP, Mapletoft RJ. 1999.** Effect of LH or GnRH on the dominant follicle of the first follicular wave in beef heifers. *Animal Reproduction Science* **57**:23–33 DOI 10.1016/S0378-4320(99)00057-3.

**Mebratu B, Fesseha H, Goa E. 2020.** Embryo transfer in cattle production and its principle and applications. *International Journal of Pharmacy & Biomedical Research* **7**:40–54 DOI 10.18782/2394-3726.1083.

**Mekonnen MM, Hoekstra AY. 2012.** A global assessment of the water footprint of farm animal products. *Ecosystems* **15**:401–415 DOI 10.1007/s10021-011-9517-8.

**Mikkola M, Hasler JF, Taponen J. 2020.** Factors affecting embryo production in superovulated Bos taurus cattle. *Reproduction, Fertility and Development* **32**:104–124 DOI 10.1071/RD19279.

**Murphy B. 2018.** Equine chorionic gonadotropin: an enigmatic but essential tool. *Animal Reproduction* **9**:223–230.

**Naderi MM, Sarvari A, Milanifar A, Boroujeni SB, Akhondi MM. 2012.** Regulations and ethical considerations in animal experiments: international laws and islamic perspectives. *Avicenna Journal of Medical Biotechnology* **4**:114.

**Naranjo-Chacón F, Montiel-Palacios F, Canseco-Sedano R, Ahuja-Aguirre C. 2019.** Embryo production in middle-aged and mature Bos taurus × Bos indicus cows induced to multiple ovulation in a tropical environment. *Tropical Animal Health and Production* **51**:2641–2644 DOI 10.1007/s11250-019-01975-2.

**Nasser L, Sá Filho M, Reis E, Rezende C, Mapletoft R, Bó G, Baruselli P. 2011.** Exogenous progesterone enhances ova and embryo quality following superstimulation of the first follicular wave in Nelore (Bos indicus) donors. *Theriogenology* **76**:320–327 DOI 10.1016/j.theriogenology.2011.02.009.

**Noseir WM. 2003.** Ovarian follicular activity and hormonal profile during estrous cycle in cows: the development of 2 versus 3 waves. *Reproductive Biology Endocrinology* **1**:1–6 DOI 10.1186/1477-7827-1-1.

**Phillips PE, Jahnke MM. 2016.** Embryo transfer (techniques, donors, and recipients). *Veterinary Clinics: Food Animal Practice* **32**:365–385.

**Pursley J, Mee M, Wiltbank M. 1995.** Synchronization of ovulation in dairy cows using PGF 2$\alpha$ and GnRH. *Theriogenology* **44**:915–923 DOI 10.1016/0093-691X(95)00279-H.

**Racowsky C, Vernon M, Mayer J, Ball GD, Behr B, Pomeroy KO, Wininger D, Gibbons W, Conaghan J, Stern JE. 2010.** Standardization of grading embryo morphology. *Journal of Assisted Reproduction and Genetics* **27**:437–439 DOI 10.1007/s10815-010-9443-2.

**Rivera FA, Mendonca LG, Lopes G, Santos JE, Perez RV, Amstalden M, Correa-Calderon A, Chebel RC. 2011.** Reduced progesterone concentration during growth of the first follicular wave affects embryo quality but has no effect on embryo survival post transfer in lactating dairy cows. *Reproduction* **141**:333 DOI 10.1530/REP-10-0375.

**Roche JR, Friggens NC, Kay JK, Fisher MW, Stafford KJ, Berry DP. 2009.** Invited review: body condition score and its association with dairy cow productivity, health, and welfare. *Journal of Dairy Science* **92**:5769–5801 DOI 10.3168/jds.2009-2431.

**Roelofs J, López-Gatius F, Hunter R, Van Eerdenburg F, Hanzen C. 2010.** When is a cow in estrus? Clinical and practical aspects. *Theriogenology* **74**:327–344 DOI 10.1016/j.theriogenology.2010.02.016.

**Selk G. 2002.** Embryo transfer in cattle, Oklahoma State University Cooperative Extension Service Fact Sheet 3158. *Available at https://shareok.org/handle/11244/49938*.

**Sheetal S, Prasad S, Gupta H. 2018.** Effect of insulin or insulin-like growth factor-I administration at mid-luteal phase of the estrous cycle during superovulation on hormonal profile of Sahiwal cows. *Veterinary World* **11**:1736.

**Shehab El-Din A, Abdel-Khalek A-K, Bakr H. 2010.** Study on follicular dynamics in superovulated cattle. MS thesis, Faculty of Agriculture, Mansoura University.

**Singh N, Dhaliwal G, Malik V, Dadarwal D, Honparkhe M, Singhal S, Brar P. 2015.** Comparison of follicular dynamics, superovulatory response, and embryo recovery

between estradiol based and conventional superstimulation protocol in buffaloes (Bubalus bubalis). *Veterinary World* **8**:983 DOI 10.14202/vetworld.2015.983-988.

**Son D-S, Choe C-Y, Cho S-R, Choi S-H, Kim H-J, Kim I-H. 2007.** The effect of reduced dose and number of treatments of FSH on superovulatory response in CIDR-treated Korean native cows. *Journal of Reproduction and Development* **53**:1299–1303 DOI 10.1262/jrd.19045.

**Tadesse M, Degefa T, Jemal J, Yohanis A, Seyum T. 2016.** Evaluation of response to super-ovulation, estrous synchronization and embryo transfer in local Zebu or crossbred dairy cattle. *Ethiopian Journal of Agricultural Sciences* **26**:27–35.

**Tahir MN, Riaz R, Bilal M, Nouman HM. 2019.** Current standing and future challenges of dairying in Pakistan: a status update. In: *Milk production, processing and marketing*. London: InTech Open DOI 10.5772/intechopen.83494.

**Thatcher W, Drost M, Savio J, Macmillan k, Entwistle K, Schmitt E, De la Sota R, Morris G. 1993.** New clinical uses of GnRH and its analogues in cattle. *Animal Reproduction Science* **33**:27–49.

**Viana JHM, Figueiredo ACS, Gonçalves RLR, Siqueira LGB. 2018.** A historical perspective of embryo-related technologies in South America. Embrapa Recursos Genéticos E Biotecnologia-Artigo Em Anais de Congresso (ALICE): Animal Reproduction, V. 15, Supl. 1. 963–970 Abstract 098.

**Warriach H, McGill D, Bush R, Wynn P, Chohan K. 2015.** A review of recent developments in buffalo reproduction—a review. *Asian-Australasian Journal of Animal Sciences* **28**:451 DOI 10.5713/ajas.14.0259.

**Wheeler MB, Monaco E, Bionaz M, Tanaka T. 2010.** The role of existing and emerging biotechnologies for livestock production: toward holism. *Acta Scientiae Veterinariae* **38**:s463–s484.

**Wiley C, Jahnke M, Redifer C, Gunn PJ, Dohlman T. 2019.** Effects of endogenous progesterone during ovarian follicle superstimulation on embryo quality and quantity in beef cows. *Theriogenology* **129**:54–60 DOI 10.1016/j.theriogenology.2019.01.024.