# Peer review of "GnRH or estradiol benzoate combination with CIDR improves in-vivo embryo production in bovines (Bos indicus and Bos taurus) under subtropics"

_PeerJ, doi:10.7717/peerj.12077_

## Round 0.1 · original submission · Major Revisions

Three expert reviewers have peer-reviewed your manuscript and all of them have made really careful and instructional comments. In addition to language issue, other major concerns were also raised and listed there. Please refer to reviewers' comments for details.

Reviewer 1 ·

Basic reporting

Dear Editor,
The present review concerns the manuscript: “GnRH or Estradiol benzoate combination with CIDR improves in-vivo embryo production as compared to conventional superovulation in bovine (Bos indicus and Bos taurus) under subtropics”. It is an interesting topic where the authors propose something new in terms of new knowledge in the area of embryo transfer.
However, in my view, the authors were unable to expose the data with sufficient clarity to approve the manuscript.
The title could be improved.
Abstract:
Insert how many animals were allocated in each group;
I couldn't find out if it was 70 cows or 90 cows?
Which breeds were crossbred cows?
The abstract was very confusing (relative to the control group), because it was difficult to know if they received FSH or not .... This has to be well specified.
Was the superovulation done in just one moment ( same time) or in several moments?
P.7- L.41: Both groups were given SO of 11o. on the 13th day. However, elsewhere, the control also received (p.7, L48-49). He's confused.
p.7 - L 49: 90 cows? It is necessary to clarify
p.7-L50-51: phrase out of context. Add in the methodology;
What is “nondescript recipient?
Ovarian structures? What is? Specify which structures?
Again -….number of animals per group?
P.7 L52: what does super estrus follicles mean?
In the abstract: results are missing or poor described;
P.7 Substantially lower: what is it?
P7. - L55: enter the p value. The p value is missing from the table as well.
P7. 56-58: Confused. To clarify
P7 - L 62-65: improving the conclusions
P7 - L77: non descript animals? What is that?
P8- L88-89: references are missing
P.9-L114: total animals (70 or 90 cows)?
P.9-L129: on on ??? what is this?
P21 - There is a lack of incomplete legends in the figures;
Give a number to the figures
P26: what kind of animal it was (figure captions)
Table: should be opened. There is no vertical line in the table
Table 1: symbols are missing from the variables (or % or “n”). Say what specie is ? (cow)
P10- L136: what is super estrus?
Table 1 - clarify better
No Abstract. embryos there is no mention of frozen embryos;
I could not observe the pregnancy rate of the transferred embryos. Place on the table. Will be better.
P 13- L 241- Reference: MS Khan, 2002 is not used.
Pregnancy rate of groups? where?
Table 1 legend needs to be improved (legend will go below)
Was the cows heat induced or were submitted to estrus synchronization for superovulation?

Mr. Editor,
The English language is not within my competence, but there are deficiencies in the English language.

Experimental design

See please, my general commentaries.

Validity of the findings

See please my general commentaries

Additional comments

Dear Editor,
The present review concerns the manuscript: “GnRH or Estradiol benzoate combination with CIDR improves in-vivo embryo production as compared to conventional superovulation in bovine (Bos indicus and Bos taurus) under subtropics”. It is an interesting topic where the authors propose something new in terms of new knowledge in the area of embryo transfer.
However, in my view, the authors were unable to expose the data with sufficient clarity to approve the manuscript.
The title could be improved.
Abstract:
Insert how many animals were allocated in each group;
I couldn't find out if it was 70 cows or 90 cows?
Which breeds were crossbred cows?
The abstract was very confusing (relative to the control group), because it was difficult to know if they received FSH or not .... This has to be well specified.
Was the superovulation done in just one moment ( same time) or in several moments?
P.7- L.41: Both groups were given SO of 11o. on the 13th day. However, elsewhere, the control also received (p.7, L48-49). He's confused.
p.7 - L 49: 90 cows? It is necessary to clarify
p.7-L50-51: phrase out of context. Add in the methodology;
What is “nondescript recipient?
Ovarian structures? What is? Specify which structures?
Again -….number of animals per group?
P.7 L52: what does super estrus follicles mean?
In the abstract: results are missing or poor described;
P.7 Substantially lower: what is it?
P7. - L55: enter the p value. The p value is missing from the table as well.
P7. 56-58: Confused. To clarify
P7 - L 62-65: improving the conclusions
P7 - L77: non descript animals? What is that?
P8- L88-89: references are missing
P.9-L114: total animals (70 or 90 cows)?
P.9-L129: on on ??? what is this?
P21 - There is a lack of incomplete legends in the figures;
Give a number to the figures
P26: what kind of animal it was (figure captions)
Table: should be opened. There is no vertical line in the table
Table 1: symbols are missing from the variables (or % or “n”). Say what specie is ? (cow)
P10- L136: what is super estrus?
Table 1 - clarify better
No Abstract. embryos there is no mention of frozen embryos;
I could not observe the pregnancy rate of the transferred embryos. Place on the table. Will be better.
P 13- L 241- Reference: MS Khan, 2002 is not used.
Pregnancy rate of groups? where?
Table 1 legend needs to be improved (legend will go below)
Was the cows heat induced or were submitted to estrus synchronization for superovulation?

Mr. Editor,
The English language is not within my competence, but there are deficiencies in the English language.

·

Basic reporting

1. BASIC REPORTING

The English language is unclear I recommend major editing by a native speaker

Figure 2 and 3 with very low resolution and the figure legends must be checked

Experimental design

The paper is in the journal scope but the novelty of paper is not 100%
Research question is not well established, the gap must be filled enough with more relevant information

Validity of the findings

The article is novel but the idea is studied many times before with the same combination
The benefit of the paper is not clearly stated
The statistically analysis must be provide in details by each test to provide the rationale of data

Additional comments

The plagiarism checker was adequate percentage

I again recommend improvement of the English language there were many grammatical errors in many lines The English language should be improved to understand your text.Some examples where the language could be improved include lines 36, 48, 131, 155 – the current phrasing makes comprehension difficult.


4. Confidential notes to the editor and authors
Special comments:
Abstract: line 38 please correct the grammatical error
Line 36: Any abbreviation must be written at first in details please check
Line 39 what do you mean by good quality corpus luteum
Line 47: authors stated (but the remaining procedure was the same…...) please clarify
Line 59-62: the conclusion is so vague and wide please adjust
Keywords: CIDR based superovulation; MOET in subtropics are not suitable words please add keywords not in the title to be more general
Introduction: first paragraph is not understood
Line 101-111 : this part need more relevant references ,,,there were many papers discussing those points
What about the main idea>>the paper hypothesis ???
Line 116: please provide references to BCS
Line 127-144: very long paragraph and filled with information
I suggest summarizing this part
Also I suggest to provide the diagram summarize the methodology and the research question to be more simple to the readers
Line 144: what about sample size ,inclusion and exclusion criteria???
Line 150-154: image must be adjusted with better resolution
Line 155: what is the ultrasound settings:::
159-166: very long paragraph rich with many grammatical error I agai suggest and English editing to this paper
Freezing and Storage of Embryos and fertility paragraphs were difficult to understand. Both showed by adjusted with many references with removing all extra spaces and correct the errors in those two paragraphs
Line 192-201: did data checked first for normality of not???
Please give the rationale for using one way ANOVA
Line 201: P value must be italic
Results
Line 216-219: where is the table of figure determine this result in return to estrous please clarify,,,,
Discussion
Please add more on application of MOET protocols…. With relevant new references
The discussion must start with similarity with others then ,,,the disagreement points for each item in the result section,,,, this section need to be rewritten again as its generalized with vague rationale
Conclusion ; add more in this part

·

Basic reporting

The English language used in this manuscript requires major improvement. It includes terms or expressions that are not technically correct. In general, many sections are difficult to understand. I recommend that the authors have their manuscript revised by a professional language editing service.
The introduction of the manuscript fails to demonstrate the contribution of the study to the field of knowledge. Relevant prior literature was not included. I suggest that the authors make an updated review of literature from the last five years to see all that has been done in the area where their study fits.
Captions for the figures and table are not clear. For example, the label for figure 3 represents only part of it.

Experimental design

The research question is not relevant. There was no knowledge gap being investigated and therefore the study did not contribute to filling any knowledge gap, as a number of previous studies have addressed the same topic. I suggest that the authors update their information.
The study included three experimental groups. However, one of them included twice as many animals as the other two; this can lead to errors in the statistical analysis.
The number of animals of each breed in each group was not included, even though the breed was considered as a factor in the analysis.

Validity of the findings

There is no clear contribution of this manuscript to the existing literature. I recommend that the authors revise the existing literature to update their knowledge on the field.

Additional comments

Please find some particular comments on your manuscript in the file attached. They're made to improve it.

---

## Round 0.2 · Minor Revisions

Both reviewers have pointed out the problem of writing and language. Authors need to carefully go through the whole manuscript for a thorough editing.

·

Basic reporting

The English language is still unclear I recommend minor editing by a native speaker

Experimental design

perfect
all suggestions are now modified

Validity of the findings

The article is novel
the idea is studied many times before with the same combination
The benefit of the paper is now clearly stated

·

Basic reporting

The manuscript revised was a corrected version of the original one. However, although most of the reviewer's comments were considered, the corrections made were not sufficient to render the manuscript suitable for publication. The introduction is very long, with confusion in the objective of the study, and with no information to support it. I'm not saying that the topic addressed is not important, but it is not clearly stated in the document. In addition, there are not sufficient references to support the study. The presentation of the results is somewhat confusing, and the discussion is repetitive and not clear. The conclusions should also be revised.
One of the main flaws I found in this manuscript is the use of the English language. Although the authors said it was checked by someone who speaks English, it is still unclear and confusing and uses many terms that are not quite technically correct. The misuse of the language makes most of the manuscript hard to understand and follow. Therefore, I recommend the authors have the manuscript checked by a professional language editing service. Maybe they could first send it to check by an English-speaking colleague to choose the most appropriate technical terms.

Experimental design

It is not clear how the research from this study fills a specific knowledge gap. The writing of the material and methods section needs improvement.

Validity of the findings

It is not clear the contribution of the results to the research area. The conclusions need revision. The study did not evaluate modified superovulation protocols, but hormonal combinations that can be used before the typical superovulation protocol.

Additional comments

Even though the authors made most of the corrections indicated and checked the language use, it was not enough to make the manuscript suitable for publication. The manuscript needs major revision, including the information considered as background for the study, the aim of the study, and the relevance of the research question. The knowledge gap being investigated is not clear, nor is the contribution of the study to filling that gap. The use of the English language requires major improvement. In my opinion, it should be first revised thoroughly by the authors, including the use of the most appropriate technical terms, and then have it checked by a professional language editing service. After doing that, you could try submitting it for a new review, particularly if you consider that the information contained in your study is important.

---

## Round 0.3 · accepted · Accept

Questions have been addressed.